# Genome-Wide Association Study of Early Vigour-Related Traits for a Rice (*Oryza sativa* L.) *japonica* Diversity Set Grown in Aerobic Conditions

**DOI:** 10.3390/biology13040261

**Published:** 2024-04-15

**Authors:** Wenliu Gong, Christopher Proud, Ricky Vinarao, Shu Fukai, Jaquie Mitchell

**Affiliations:** School of Agriculture and Food Sustainability, The University of Queensland, Brisbane, QLD 4072, Australiajaquie.mitchell@uq.edu.au (J.M.)

**Keywords:** GWAS, early vigour, mesocotyl length, aerobic rice, direct seeding

## Abstract

**Simple Summary:**

In this study, we explored a relatively new system of growing rice called aerobic rice production, where rice is direct-seeded and grown with sufficient water, rather than being flooded. Early vigour-related traits are important in aerobic conditions. Our main goal was to find quantitative trait loci and potential genes related to the plant’s early growth and ability in the water-saving conditions. By conducting experiments in both fields and glasshouses, we discovered significant differences in growth among different rice varieties and identified 32 quantitative trait loci linked to these early vigour-related traits. Two quantitative trait loci were found to be crucial and connected to 23 potential growth and stress response genes. These findings highlighted the genetic basis of early vigour-related traits under aerobic conditions and suggest pathways for breeding better-performing rice varieties through marker-assisted selection.

**Abstract:**

Aerobic rice production is a relatively new system in which rice is direct-seeded and grown in non-flooded but well-watered conditions to improve water productivity. Early vigour-related traits are likely to be important in aerobic conditions. This study aimed to identify quantitative trait loci (QTL) and candidate genes associated with early vigour-related traits in aerobic conditions using a *japonica* rice diversity set. Field experiments and glasshouse experiments conducted under aerobic conditions revealed significant genotypic variation in early vigour-related traits. Genome-wide association analysis identified 32 QTL associated with early vigour-related traits. Notably, two QTL, *qAEV1.5* and *qAEV8*, associated with both early vigour score and mesocotyl length, explained up to 22.1% of the phenotypic variance. In total, 23 candidate genes related to plant growth development and abiotic stress response were identified in the two regions. This study provides novel insights into the genetic basis of early vigour under aerobic conditions. Validation of identified QTL and candidate genes in different genetic backgrounds is crucial for future studies. Moreover, testing the effect of QTL on yield under different environments would be valuable. After validation, these QTL and genes can be considered for developing markers in marker-assisted selection for aerobic rice production.

## 1. Introduction

Rice is traditionally grown under flooded conditions to maximize yield in the Riverina production environment in Australia. Over the past few decades, drill seeding has become increasingly popular and accounted for 70% of Australia’s sowing in the 2018–2019 season [1]. With ever-increasing water restriction, aerobic rice in which rice is grown in well-drained, non-flooded and unsaturated, but well-watered conditions, has been suggested to improve water productivity [2]. While the shift to aerobic rice production could bring substantial advantages, particularly in water savings, there are some challenges, particularly in relation to direct-seeded aerobic rice. For example, Chamara et al. [3] found that crop emergence decreased 60% for direct seeding at a 2 cm sowing depth compared to a 0.5 cm sowing depth in the Philippines. Poor crop establishment was the primary factor leading to a low and unstable yield of direct-seeded aerobic rice [4]. Additionally, weed growth can pose a significant challenge to aerobic production due to absence of the water layer [5]. In order to enhance rice productivity in direct-seeded aerobic rice systems, rice genotypes with high early vigour are likely to be required.

Early vigour is defined as the plant’s capacity to grow rapidly during the seedling stage and has been shown to play a crucial role in determining the ability of seeds to emerge quickly and lead to an ideal field establishment [6]. Early vigour is an agronomic trait that integrates both the plant genetics, physiology and the environment. Namuco et al. [7] reported that increasing early vigour decreased the influence of weed competition. In rice, early vigour is typically associated with germination rate, plant height, shoot biomass, leaf area, seed weight and mesocotyl elongation [8,9,10,11]. Rapid and uniform seed germination is essential for efficient resource utilization during the early growth stage, allowing seedlings to acquire nutrients and water from the soil quickly [8,9]. Plants that are taller and have greater biomass and leaf area can acquire more light and resources from the environment, leading to rapid growth and early canopy closure [10,11]. Additionally, if drill sowing (recommended 2.5–4 cm depth [12]) is adopted in direct seeding to prevent bird damage and lodging after the heading stage [13], mesocotyl elongation becomes critical for good establishment. Mesocotyl elongation can facilitate the rapid emergence of seedlings from the soil surface, leading to stronger and more uniform seedling establishment [13,14]. Ohno et al. [15] found that genotypes with longer mesocotyl had a higher rate of emergence (r = 0.67 **) and shorter days to emergence (r = −0.79 **) among 16 rice genotypes under field conditions planted at an 8.5 cm depth for the screening of mesocotyl elongation. Evaluation of the significance of early vigour-related traits for aerobic production is required, and exploring the underlying genetics that influence the key early vigour traits can provide benefit for genetic improvement for the development of germplasm adapted to aerobic production.

Using linkage mapping methods, studies have been conducted in rice to evaluate the genomic regions associated with early vigour-related traits [16,17,18,19] and mesocotyl length [14,20,21]. For example, utilizing a recombinant inbred line population derived from Lemont and Teqing, Zhang et al. [16] found 18 quantitative trait loci (QTL) associated with early vigour-related traits (plant height, dry weight and seed weight) in pot experiments. In the study of Lee et al. [20], five QTL were identified to be associated with mesocotyl length using a backcross inbred line population from Kasalath and Nipponbare in plastic jar experiments. A genome-wide association study (GWAS) has also been conducted to evaluate the genomic regions in rice for early vigour [22,23,24] and mesocotyl length [25,26,27]. Although the correlations between mesocotyl length and early growth traits and related genomic regions were reported, whether there are common genomic regions and candidate genes associated with both early vigour-related traits and mesocotyl length under aerobic field conditions remains unknown.

In this study, GWAS was performed for early vigour-related traits utilizing a *japonica* diversity set grown under aerobic field and glasshouse conditions. The objectives were to (1) investigate the genotypic variation for early vigour-related traits in a *japonica* diversity set and explore the relationships among traits; (2) identify genomic regions associated with early vigour-related traits in rice grown under aerobic conditions; and (3) identify potential candidate genes involved in early vigour under aerobic conditions.

## 2. Materials and Methods

### 2.1. Experiment Location

Two glasshouse experiments and three field experiments were conducted at the University of Queensland, Australia, in this study. The first glasshouse experiment (GH18) was conducted from November to December 2018 in a temperature-controlled glasshouse at Gatton Campus (27.5551° S, 152.3369° E). The second glasshouse experiment (GH21) was conducted from November to December 2021 in a temperature-controlled glasshouse at St Lucia Campus (27.4975° S, 153.0137° E). The three aerobic field experiments were conducted in three seasons, 2018–2019 (FIELD19), 2019–2020 (FIELD20) and 2021–2022 (FIELD22), at Gatton Campus.

### 2.2. Plant Material and Experimental Design

A diverse set of 302 genotypes, which was a mix of lowland and upland *japonica* genotypes compiled by New South Wales Department of Primary Industries, was used in this study. There were 74, 68, 14, 81, 60 and 5 genotypes from Australia, America, Africa, Asia, Europe and unknown sources, respectively. Of these, 302, 292, 283, 241 and 286 genotypes were examined in GH18, GH21, FIELD19, FIELD20 andFIELD22, respectively. The package *DiGGer* [28] was used to generate a partially replicated row–column design in GH18, FIELD19 and FIELD20. The R package *ODW* [29] was used to generate a partially replicated row–column design in GH21 and FIELD22.

### 2.3. Cultural Details of Glasshouse Experiments

In GH18, a balance (Mettler Toledo Analytical balance, 4 decimal places, Melbourne, VIC, Australia) was used to weigh four individual seeds that weighed within 2 mg of the average paddy weight, and the seeds of each genotype were put in an Eppendorf tube in preparation for planting. The tube pots (123 mm height × base of 50 mm square bottomless forestry tubes; 0.21 L; https://www.gardencityplastics.com/t50sfk.html, accessed on 1 September 2018) were lined with cotton to prevent soil escape. The pots were filled with 5 cm of Vertosol soil (a large water-holding capacity soil type with brown or black colour with clay textures, Isbell and NCST [30]) from the Crop Research Unit (27.5393° S, 152.3330° E), Gatton, QLD, Australia. Two seeds were sown at 7 cm from the soil surface to determine genotypic potential of mesocotyl length. Replications one and two were planted on the 12 and 13 November 2018, respectively.

In GH21, one-litre ANOVA^®^ pots (http://www.anovapot.com/, accessed on 1 September 2020) filled with 1.20 ± 0.05 kg Ferrosol soil (a well-drained soil type with red colour with clay–loam to clay textures, Isbell and NCST [30]) from the Redlands Research Station (27.5271° S, 153.2505° E), Cleveland, QLD, Australia, were used for planting. An inverted 5 cm Petri dish was placed 5 cm from the base of the pot to reduce root escape, so any effects of the roots were nullified in the set-up. The Petri dishes were covered with a thin layer of soil, and then 6 g of slow-release Osmocote^®^ Pro 5–6 M fertilizer (17N-11P-10K + 2MgO + TE, ICL Specialty Fertilizers) was put on top of the thin soil layer, and the pots were filled to the specific weight. Six seeds were sown at 2.5 cm depth below the soil surface on 21 November 2021, and one plant was retained in each pot 10 days after sowing (DAS). The pots were placed onto a capillary mat for maintenance of the water supply [31]. Adapting the system described by Hunter et al. [32], the aerobic environment was established by maintaining a constant water table at 4 cm to the base of the pot with a float valve, and all pots were connected to the same reservoir.

### 2.4. Field Experiments Management

In the three field experiments, the plots were 2 m in length and consisted of 7 rows with an inter-row spacing of 0.22 m. In FIELD19, a basal fertilizer was used at a rate of 480 kg ha^−1^ at 5 days before sowing using Incitec Pivot’s Pastureboosta (24-4-13-4) with 15 kg ha^−1^ of zinc sulphate monohydrate included. A pre-emergent herbicide treatment was applied at 5 days after the first irrigation. The treatment used standard rates of the following: (1) Stomp 440 (440 g L^−1^ Pendimethalin, CropCare, Murarrie, QLD, Australia), (2) Magister (480 gL^−1^ Clomazone; FMC Australia, North Ryde, NSW, Australia) and (3) Gramoxone (250 g L^−1^ Paraquat, Syngenta, Basel, Switzerland). In FIELD20 and FIELD22, a basal fertilizer was applied at a rate of 400 kg ha^−1^ using Incitec Pivot’s CK140S (23-2-18-4) at 5 days before sowing, along with 15 kg ha^−1^ of zinc sulphate monohydrate. The pre-emergent herbicide treatment in FIELD20 and FIELD22 was consistent with that of FIELD19.

Seeds were drill-sown at a rate of 90 kg ha^−1^ on the 30 November 2018 in FIELD19, 130 kg ha^−1^ on the 16 October 2019 in FIELD20 and 130 kg ha^−1^ on the 8 October 2021 in FIELD22. Seeds were sown at 3–4 cm depth. Irrigation by a lateral boom irrigator was applied three times a week in all the field experiments.

### 2.5. Measurements of Traits

#### 2.5.1. Glasshouse Traits Measurements

In GH18 and GH21, the pots were checked daily, and emergence was recorded when coleoptile broke the soil surface. The plant height (cm) from the soil to the tip of the youngest fully emerged leaf was collected before harvest at 19 and 22 DAS for GH18 and GH21, respectively. At harvest (19 DAS), in GH18, mesocotyl length (mm) was measured by Vernier Calliper. The weight of the total biomass (aboveground, mg) was recorded after the plants were dried in the oven (60 °C) for three days.

#### 2.5.2. Field Traits Measurements

In the three field experiments, the early vigour score was rated on a 1–9 scale, where 1 represented the most vigorous and 9 represented the least vigorous following Standard Evaluation System for Rice [33]. Early vigour score was taken at 30 DAS in FIELD19, 35 DAS in FIELD20 and 30 DAS in FIELD22.

In FIELD20, early vigour harvest took place at 39 DAS, when two representative 33 cm rows were selected randomly by placing a 33 cm ruler between rows and seedlings were cut from the base of the shoot. Two plants from each plot were randomly chosen from the field, and a ruler was used to measure the early plant height (cm) measured from the ground to the leaf apex at 40 DAS. The seedlings were in an oven for 3 days (65 °C), and then the total biomass was determined using a balance. The total biomass (aboveground, g m^−2^) was calculated from the seedling dry weight (g) per area harvested (m^2^). The light interception measurements were recorded at 47 DAS. The photosynthetically active radiation (PAR) was measured by placing a ceptometer (AccuPAR, Decagon Devices Inc., Pullman, WA, USA) at a 45° angle to the rows between 11 a.m. and 1 p.m. Measurements were recorded above (Ra) and immediately below the canopy (Ru). The light interception efficiency was determined by using the fraction of PAR incepted. Light interception efficiency = (Ra − Ru)/Ra × 100%.

### 2.6. Phenotypic Data Analysis

A multiplicative linear mixed model was used for analysis and was implemented in *ASReml-R* in the R environment (V4.2) [34]. The best spatial model was fitted for each trait using the method described by Isik et al. [35]. Genotype was treated as both a random effect to estimate heritability and a fixed effect to obtain the best linear unbiased estimates (BLUEs), while the blocking terms (replication, row and column) were treated as random effects.

Heritability refers to the proportion of phenotypic variation in a population that is attributable to genetic variation and was calculated using Cullis_H_2_ (Calculate Generalized Heritability from lme4 Model) package.
Heritability = 1 − (vblup/(2 × var_g)).
vblup—the average standard error of differences between BLUPs squared.var_g—the genotypic variance.

Principal component analysis (PCA) was carried out via singular value decomposition on the standardized and centred BLUEs for each environment. A PCA biplot was generated using three packages, *tidyverse* [36], *ggrepel* [37] and *ggpubr* [38], implemented in the R environment (V4.2). The violin plots of different origin groups were generated using *ggpubr* package. The comparisons among origin groups were carried out using *t*-test at *p* < 0.05.

### 2.7. SNP Genotyping

Single nucleotide polymorphism (SNP) genotyping and filtering were similar to the methods described by Vinarao et al. [39], with some modifications. A service provider (Diversity Arrays Technology, Bruce, SA, Australia) carried out SNP genotyping including deoxyribonucleic acid (DNA) extraction. Briefly, the SNP data were obtained through a genotyping-by-sequencing approach using the diversity arrays technology sequencing platform (DArTSeq). DNA libraries were sequenced using a next-generation sequencing platform (HiSeq 2500, Illumina, Melbourne, VIC, Australia), and the resulting sequences were aligned to the *Oryza sativa* v7.0 reference genome (https://jgi.doe.gov/, accessed on 1 September 2020) to identify SNPs. SNP genotype data obtained from DArTSeq were processed and quality controlled using the *dartr* package [40] in the R environment (V4.2). In brief, quality control and filtering for high-quality SNPs included (1) removal of secondaries—the occurrence of multiple SNP loci within a fragment, (2) retention of SNP with call rate ≥ 0.80 and (3) filtering out of SNPs with minor allele frequency (MAF) of <0.05. The physical base pair position of the first nucleotide in the tagged sequences was used as map positions. Missing SNP genotype information was imputed using random forest imputation of 500 trees carried out in *missForest* v1.4 [41].

### 2.8. Genome-Wide Association Study

The linkage disequilibrium (LD) decay between the SNP markers was analysed using squared allele frequency correlation (r^2^) values with significant *p* values <0.1 for each pair of loci. The physical distance, which reduced r^2^ to half of the maximum value, was used as the LD decay rate [42].

The GWASs were conducted by the Genome Association and Prediction Integrated Tool (GAPIT) package in the R environment (V4.2) [43]. Bayesian-information and Linkage-disequilibrium Iteratively Nested Keyway (BLINK) was used to conduct GWAS implemented in GAPIT [44]. Previously, the general linear model and mixed linear model were the major analysis models for GWAS; however, the computation speed of those models is low. In the fixed and random model circulating probability unification model, the iterative association of the fixed effect and random effect models not only can handle large sample size but also can detect massive density markers [45]. The BLINK model not only accelerates computational processes but also demonstrates superior statistical power compared to alternative models [44]. The false discovery rate <0.05 for SNPs was set as the threshold to be considered significant [46,47,48].

### 2.9. QTL Nomenclature and Identification of Candidate Genes

Significantly associated SNPs occurring within 200 kb of each other were grouped and classified as one QTL, and the value of 200 kb was chosen based on previous estimates of global LD in the rice in general [42,49,50,51]. QTL nomenclature was carried out following the guidelines set by the Gene Nomenclature System for Rice [52]. The QTL names were made *qAEV* for genomic regions associated with early vigour-related traits in aerobic conditions.

Post QTL analysis was carried out by identification of candidate genes in the promising genomic regions identified in the current study. A total of two QTL associated with early vigour score and mesocotyl length were prioritized for candidate gene analysis. Predicted candidate genes within 200 kb upstream and downstream, similar to previously reported analyses [53,54,55], of each significantly associated QTL were identified using the genome browser of the Rice Genome Annotation Project [56].

## 3. Results

### 3.1. Phenotypic Variation of Traits

The average mesocotyl length was 9.7 mm, with the longest genotype Labelle at 37.2 mm when seed was sown at 7 cm depth (Table 1). Most genotypes in the diversity set had a mesocotyl length less than 15 mm (254 genotypes, 84%). The early vigour score ranged from 0.9 to 9.2, 1.7 to 9.9 and 0.8 to 8.2 in FIELD19, FIELD20 and FIELD22, respectively. There were highly significant (*p* < 0.01) genotypic differences for mesocotyl length, early vigour score and all other early vigour-related traits recorded. Heritability of early vigour-related traits ranged from 0.40 to 0.82. High heritability was particularly observed for mesocotyl length in GH18, early vigour score in FIELD20 and plant height across three experiments.

PCA revealed that the first and second principal components accounted for 48.0% and 10.9% of the total variation (Figure 1). The early vigour score of three field experiments was significantly correlated (r = 0.45 ** to 0.54 **, Appendix A). More vigorous genotypes had shorter days to emergence and higher light interception, plant height and biomass in the field. Additionally, genotypes with longer mesocotyl length were taller, had greater biomass and fewer days to emergence in the glasshouse. Mesocotyl length in the glasshouse experiment was negatively correlated with early vigour score in the field (r = −0.35 ** to −0.38 **).

The rice genotypes from different origins were compared for mesocotyl length and early vigour score (Figure 2). Compared to Australian genotypes with a mean at 7.9 mm, the American and European genotypes had significantly longer mesocotyl lengths with means at 11.8 mm and 11.0 mm, respectively (Figure 2A). The Australian genotypes were the least vigorous among the origins (Figure 2B–D).

### 3.2. Population Structure

The number of SNPs ranged from 322 in chromosome 9 to 1084 in chromosome 1, with an average of 584 markers. The SNP marker density was 49 kb on average (Figure 3). LD dropped to half of its initial value at 105 kb in the current germplasm (Appendix A). With the average inter-marker distance of 49 kb, such marker density provides reasonable power to identify genetic variants associated with target traits, even though the causal variant was not queried.

In the PCA of the genotype data, the principal components 1, 2 and 3 accounted for 24.88%, 10.78% and 4.75%, respectively (Figure 4A). The result of PCA with separation based on geographic location is presented in Figure 4B.

### 3.3. GWAS

A total of 35 SNPs associated with early vigour and related traits were detected across five experiments, and in total 32 QTL were identified (Table 2). The SNPs are located in all chromosomes except chromosomes 10 and 12. The minor allele frequencies (MAF) of those SNPs were from 0.06 to 0.48. In total, there were 18, 5, 2, 13 and 5 SNPs in GH18, GH21, FIELD19, FIELD20 and FIELD22, respectively. The phenotypic variance explained (PVE) of each SNP was from 1.33% to 42.02%. The SNP with the highest PVE was SNP 7_24657049, which was associated with days to emergence in GH21.

It was noticeable that two QTL, *qAEV1.5* and *qAEV8*, were both significantly associated with mesocotyl length in GH18 and early vigour score in FIELD20 (Figure 5). *qAEV1.5* explained 8.59% and 9.98% of the phenotype variance for mesocotyl length and early vigour score, respectively. *qAEV8* explained 1.53%, 7.91% and 22.05% of the phenotype variance for mesocotyl length, early vigour score and biomass m^−2^ in FIELD20, respectively. Additionally, on chromosome 1, SNP 1_32510801 was significantly associated with early vigour score in FIELD19 and plant height in PH18. The SNP 1_32510801 was also co-located with SNP 1_32499722, which was associated with early vigour score in FIELD22. These two SNPs were classified within one QTL *qAEV1.2*. The QTL on chromosome 1 *qAEV1.3*, including SNP 1_34429355 and SNP 1_34600584, were significantly associated with days to emergence in FIELD20 and plant height in GH18. The SNP 1_38847713 on chromosome 1 associated with plant height was repeatedly detected in the two experiments GH21 and FIELD20. On chromosome 3, the SNP 3_1248008 was significantly associated with plant height and seedling biomass in GH18. The SNP 4_12046452 was detected to be associated with light interception in FIELD20 and plant height in GH21. The SNP with the second highest PVE at 36.33% was SNP 6_21702460, which was associated with plant height, and the SNP was also associated with early vigour score in FIELD22. In total, eight QTL, *qAEV1.2*, *qAEV1.3*, *qAEV1.5*, *qAEV1.6*, *qAEV3.1*, *qAEV4.3*, *qAEV6.1* and *qAEV8*, were detected in multiple traits across different experiments. The remaining QTL were identified for only one trait in one experiment.

There were 8 QTL associated with mesocotyl length in GH18. On average, Australian genotypes had 3.6 positive alleles that promoted mesocotyl elongation, which was significantly lower than American, African and European genotypes (Figure 6A). In both FIELD20 (Figure 6C) and FIELD22 (Figure 6D), the American, African and European genotypes had significantly more positive alleles associated with early vigour score compared to Australian genotypes.

### 3.4. Identification of Candidate Genes

The two QTL *qAEV1.5* and *qAEV8* were found to be consistently associated with both mesocotyl length and early vigour score and were further dissected for identification of candidate genes.

Within the *qAEV1.5* region, multiple genes were identified with distinct functional roles (Table 3). *LOC_Os01g62460* has sequence-specific DNA-binding transcription factor activity and has potential roles in plant growth and development. *LOC_Os01g62480* and *LOC_Os01g62600* are both laccase precursor proteins, and the two genes have implications in morphological traits (xylem structure, cell wall and cell length) and leaf development under direct-sown and drought conditions, respectively. *LOC_Os01g62500* is the *OsFtsH3* FtsH protease, which targets mitochondria and is involved in seed germination through arginine metabolism. *LOC_Os01g62570* encodes a putative ATP/GTP/Ca++ binding protein and is implicated in cell growth and post-embryonic development. The homeodomain protein encoded by *LOC_Os01g62920* delves into plant hormone regulations. *LOC_Os01g62610* encodes a peptidyl-prolyl cis–trans isomerase of the FKBP type. FKBP gene is involved in hormone signalling, plant growth and development and stress response. The gene *LOC_Os01g62800* encodes methyltransferase, and the methyltransferase gene family is related to the seed vigour index. *LOC_Os01g62810* and *LOC_Os01g62840* emphasize cellular processes and growth regulations. *LOC_Os01g62514*, *LOC_Os01g62660*, *LOC_Os01g62630*, *LOC_Os01g62900*, *LOC_Os01g62760* and *LOC_Os01g63010* are involved in drought response in plants.

Within *qAEV8*, *LOC_Os08g16260* and *LOC_Os08g16320* both encode for a putative cytochrome P450 protein. Cytochrome P450 proteins are associated with the regulation of cell size in the embryo and the process of apoptosis in the endosperm. *LOC_Os08g16480* encodes for a protein containing an ATPase. ATPase has roles associated with germination percentage and seedling growth, particularly affecting root and shoot length. *LOC_Os08g16600* encodes for a putative WD-40 repeat protein. The WD-40 gene family is known for the roles in signal transduction and the regulation of hormone-controlled plant cell division. *LOC_Os08g16570* expresses a gene of notable importance in drought resistance.

## 4. Discussion

### 4.1. Genotypic Variation and Relationship of Traits

The success of cultivating direct-seeded aerobic rice is largely dependent on the early stages of plant growth, with mesocotyl length and early vigour being key factors. The current study examined a *japonica* diversity set in field and glasshouse conditions. The genotypic variation among the diversity set was high for early vigour-related traits, while significant relationships existed among those traits. Several studies have also observed a highly significant genotypic difference in early vigour traits. Redona and Mackill [76] found highly significant differences in emergence index (a scale that favours both early emergence and a high percentage emergence), early vigour score and shoot weight at 35 DAS among 27 rice varieties in the field, with the field vigour score highly correlated with the emergence index (r = 0.50**) and dry weight (r = 0.59**) of direct-seeded rice. According to Krishna et al. [77], who evaluated early seedling vigour of dry direct-seeded rice 30 DAS in the field, there was significant variation for plant height, biomass and vigour index, and there were significant positive correlations between vigour index in relation to plant height and total biomass (r = 0.44**, r = 0.50**). The current study was one of the foremost in utilizing a *japonica* diversity set in the aerobic rice system. The large variation among different traits could be exploited further for the genetic studies for the diversity set. Additionally, significant correlations were indicative that there could be the potential for common QTL controlling multiple traits in aerobic conditions. By utilizing a diverse panel in this study, the genetic architecture of early vigour-related traits was further explored.

### 4.2. GWAS Analysis

There were two important QTL associated with both early vigour score and mesocotyl length under aerobic conditions in the current study. Of the two QTL, *qAEV1.5* was found to overlap with the reported *qSV1d*, which was associated with seedling vigour traits such as plant height and aboveground dry weight utilizing the 3K Rice Genome Project under flooded field conditions [24], and *qSV1d* shared overlapping regions with two other QTL, *qAEV1.3* and *qAEV1.4*, identified in the current study. The other important QTL *qAEV8* was collocated with *QFML8-1*, which was associated with mesocotyl length utilizing the 3K Rice Genome Project under field conditions [27]. Although GWAS had been used by others to determine the genomic regions associated with early vigour and mesocotyl length in rice, this study was one of the first to identify the QTL associated with both early vigour score and mesocotyl length, specifically under aerobic conditions and direct-drill flooded system.

Among the other identified QTL, *qAEV1.6*, which was associated with plant height in two experiments, was collocated with the reported *qPH-1b*, which was associated with plant height [78], and *qAEV1.6* was only 400 kb in distance from the *sd-1* gene, which was a pivotal gene inducing short plant height and high harvest index in rice [79]. The QTL *qAEV1.8* associated with early vigour score on chromosome 1 coincided with *qSV1e*, which was associated with plant height and aboveground dry weight in the field experiments [24]. *qAEV3.1* is near OsMADS50, which has been previously shown to have an effect on plant height in rice [80]. Both *qAEV3.3* and *qAEV4.2* were associated with mesocotyl length and were found to have less than 200 kb of distance from the two SNPs in Wu et al.’s [25] study, which were associated with mesocotyl length SNP315010911 on chromosome 3 and SNP409772335 on chromosome 4, respectively. Two QTL on chromosome 6, *qAEV6.1* and *qAEV6.2*, that were associated with plant height and early vigour score both overlapped with *qPH-6*, which was associated with plant height using a set of recombinant inbred sister lines from Nei2B and Zhonghui8006 [78].

The genetic relationships among the *japonica* diversity set revealed that, in general, compared to Australian genotypes, American genotypes had greater mesocotyl elongation and early vigour, and the number of favourable alleles of related QTL were also significantly higher. This indicates the potential influence of specific QTL in enhancing the early vigour-associated traits in American genotypes, making them viable candidates for integration into the breeding programs. It should also be noted that Australian genotypes had moderate frequency of favourable alleles for mesocotyl length and early vigour in the diversity set, and donors could be chosen directly using marker-assisted forward breeding after the establishment of an accurate marker system and validation in the target environment [81].

Genotypes with longer mesocotyl length tended to be more vigorous in both glasshouse and field conditions, which was supported by previous studies [15,82]. However, high-yielding semi-dwarf genotypes tend to have poor establishment under direct seeding, as the semi-dwarfing gene *sd-1* has been shown to have a detrimental effect on the expression of plant height and mesocotyl elongation [83]. The lack of *sd-1* in rice was associated with a loss in yield, as illustrated by Yoshida et al. [84] and Mathan et al. [85]. Conversely, taller rice genotypes could make rice more competitive in weedy upland conditions and have a better root system, which ensures the plants’ access to essential soil nutrients and water [86]. This relationship between plant height, mesocotyl length and the *sd-1* gene underlines the complexity of breeding efforts to optimize yield while ensuring adaptability and resilience to varied planting conditions. Balancing these traits becomes important in developing varieties that can reconcile the need for high yield with the adaptative traits necessary for success in direct-seeding aerobic environments. It was noted that the distance between *sd-1* and *qAEV1.5* was around 2 Mb, and after validation there is a possibility of pyramiding favourable alleles for mesocotyl elongation and the semi-dwarf gene where aerobic direct-seeding production is the target environment.

### 4.3. Candidate Genes

For the two important QTL, *qAEV1.5* and *qAEV8*, this study has identified several significant candidate genes within the detected QTL, each contributing distinct roles to plant growth and development, especially under direct-sown aerobic environments. *LOC_Os01g62460* (ZOS1-16-C_2_H_2_ zinc finger protein) and *LOC_Os01g62514* (WRKY56) are sequence-specific DNA-binding transcription factors. Both the C_2_H_2_ zinc finger proteins and WRKY gene family are pivotal in plant growth and development and are implicated in drought tolerance adaptation and the enhancement of root thickness, which were critical for optimized performance under more challenging conditions such as the aerobic production system [26,60]. The revelation of laccase precursor proteins, *LOC_Os01g62480* and *LOC_Os01g62600*, induces cellular and structural adaptations in response to environmental variation, potentially guiding enhancements in plant resilience and adaptability under direct-sown and drought conditions [58,61]. The FKBP-type gene *LOC_Os01g62610* is reported to be indicative of the regulation of hormone signalling essential for plant growth, development, stress response and seed germination [62]. The two candidate genes *LOC_Os01g62660* (MYB family transcription factor) and *LOC_Os01g62920* (homeodomain protein) are both implicated in drought response, with the latter being particularly involved in plant hormone regulations including auxin, ABA, cytokinin and GA, which were crucial in the developmental processes [23,64].

Additionally, the examination of the *qAEV8* candidate gene underscores the potential significance of cytochrome P450 proteins, encoded by *LOC_Os08g16260* and *LOC_Os08g16320*, in mediating cellular processes within the embryo and directing apoptosis in the endosperm [73]. The ATPases encoded by *LOC_Os08g16480* have an implication in seedling growth and germination percentage, particularly in root and shoot lengths, and emerge as focal points for understanding the initial establishment of plants and subsequent growth dynamics [74]. *LOC_Os08g16600* (WD-40 repeat protein family) is implicated in signal transduction and hormone-controlled plant cell division, which are central processes in the regulation of plant growth and development [68].

In order to understand the potential impact of the identified QTL and candidate genes on aerobic rice production, further studies could be conducted to assess the effect on important traits such as yield and grain quality. This validation would provide insight into whether yield and quality traits could be improved by targeting the identified QTL and candidate genes. Additionally, validation of the identified QTL and candidate genes in other genetic backgrounds is crucial to ensure the stability and reliability before adopting it into rice breeding programs. Once validated, these putative QTL and candidate genes could be used to select donors with favourable alleles for early vigour and related traits and can also be used for mark-assisted selection for an aerobic rice breeding program.

## 5. Conclusions

In conclusion, this study quantified the existence of genetic variation in early vigour-related traits, and this information can now be exploited by the Australian rice breeding program to improve plant establishment, particularly from depths using deep sowing, under direct seeding. Through genome-wide association analysis, this study identified two important QTL, *qAEV1.5* and *qAEV8*, which were both associated with early vigour score and mesocotyl length in rice. Between the two highlighted QTL, a total of 23 candidate genes were identified, most of which have been previously reported to be related to developmental regulation and stress adaptational mechanisms in various environmental conditions, further supporting their potential involvement in the adaption to direct seeding under aerobic conditions. The existence of genotypic variation and genomic regions for early vigour-related traits suggests that there is scope within the breeding program to improve the establishment of drill-direct-seeded aerobic rice, particularly when sown at depth, by exploiting the genotypes and QTL related to longer mesocotyl and greater early vigour. Further exploration into the relationship of plant height, *sd-1* gene, early vigour-related traits and mesocotyl length becomes essential for potential pyramiding of the favourable alleles after validation in target production environments.

## Figures and Tables

**Figure 1 biology-13-00261-f001:**
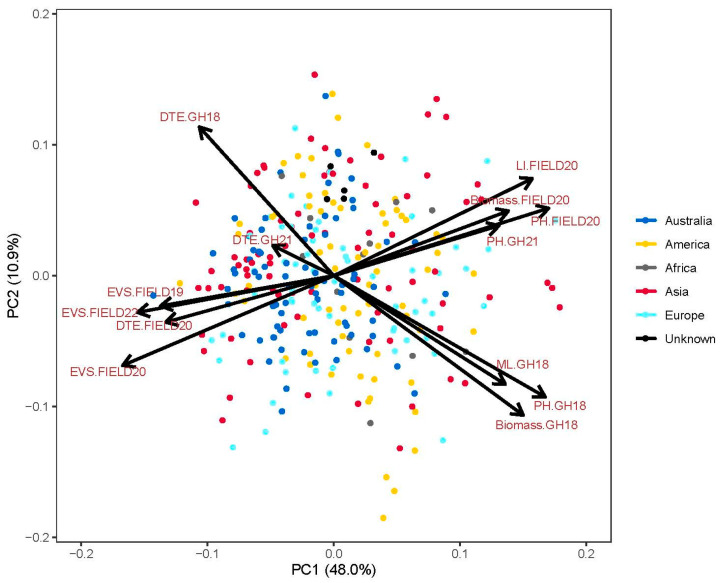
Principal component analysis biplot for the first two principal components for mesocotyl length (ML), early vigour score (EVS), plant height (PH), days to emergence (DTE), biomass and light interception (LI) in two glasshouse experiments (GH18 and GH21) and three field experiments (FIELD19, FIELD20 and FIELD22), highlighting the origins in different colours.

**Figure 2 biology-13-00261-f002:**
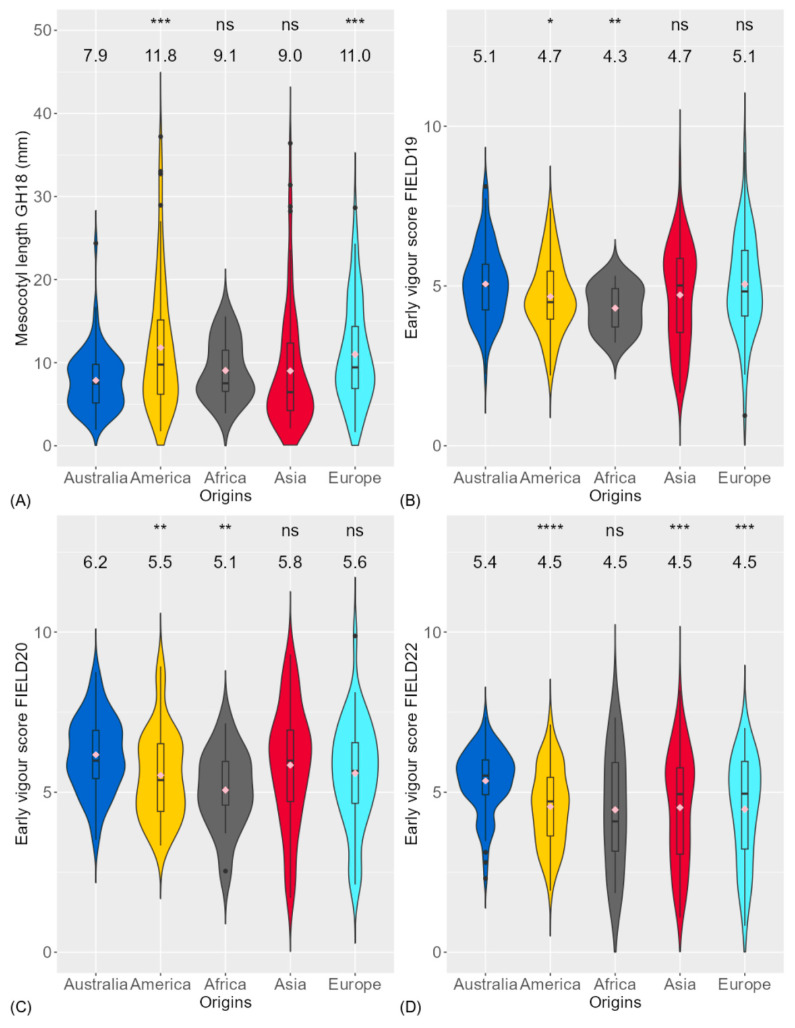
Comparison of genotypic origins for (**A**) mesocotyl length in glasshouse experiment (GH18) and early vigour score (1 is most vigorous) in field experiment in (**B**) season 2018–2019 (FIELD19), (**C**) season 2019–2020 (FIELD20) and (**D**) season 2021–2022 (FIELD22) between Australia and different origins. Significance levels between Australian genotypes and other origins were displayed on the top of each figure: ns, not significant; *, *p* < 0.05; **, *p* < 0.01; ***, *p* < 0.001; ****, *p* < 0.0001. The mean of each trait for each origin is displayed below the significance level. The diamonds within each boxplot are the mean of each trait for each region.

**Figure 3 biology-13-00261-f003:**
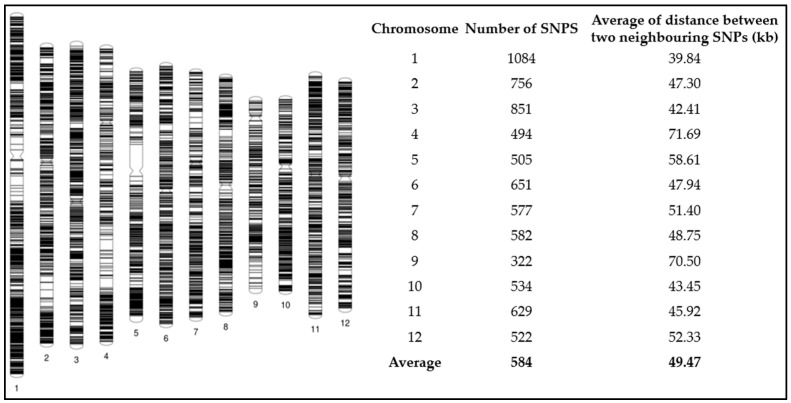
Graphical genotype of 7507 SNP markers detected in the *japonica* rice diversity set.

**Figure 4 biology-13-00261-f004:**
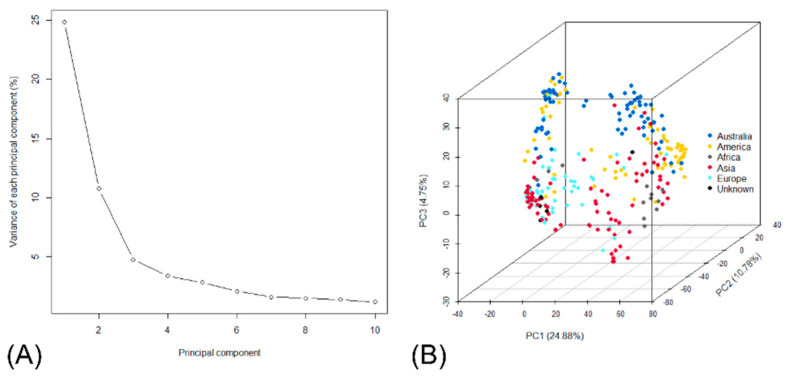
(**A**) Total variance of each principal component of genotype data. (**B**) The three principal components (PCs) capturing the variation in genotypes in the *japonica* diversity set in a principal components analysis 3D plot of genotype data. Colours represent origin of genetic material.

**Figure 5 biology-13-00261-f005:**
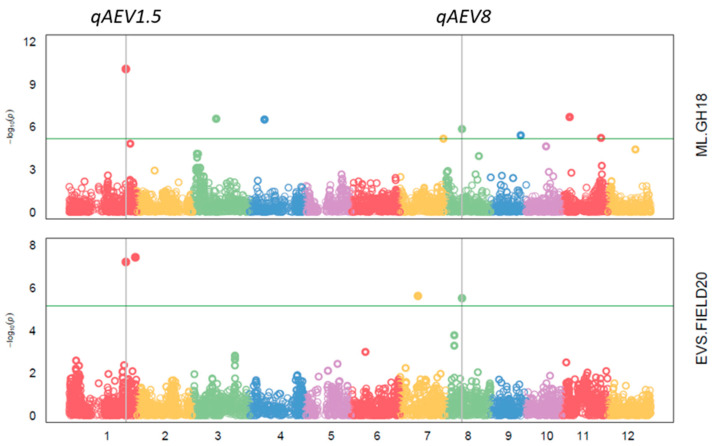
The Manhattan plot of association study of mesocotyl length (ML) in an aerobic glasshouse experiment (GH18) and early vigour score (EVS) in an aerobic field experiment (FIELD20). *qAEV1.5* and *qAEV8* were two quantitative trait loci (QTL) on chromosome 1 and chromosome 8 for early vigour traits. Green line represents threshold of false discovery rate *p* < 0.05.

**Figure 6 biology-13-00261-f006:**
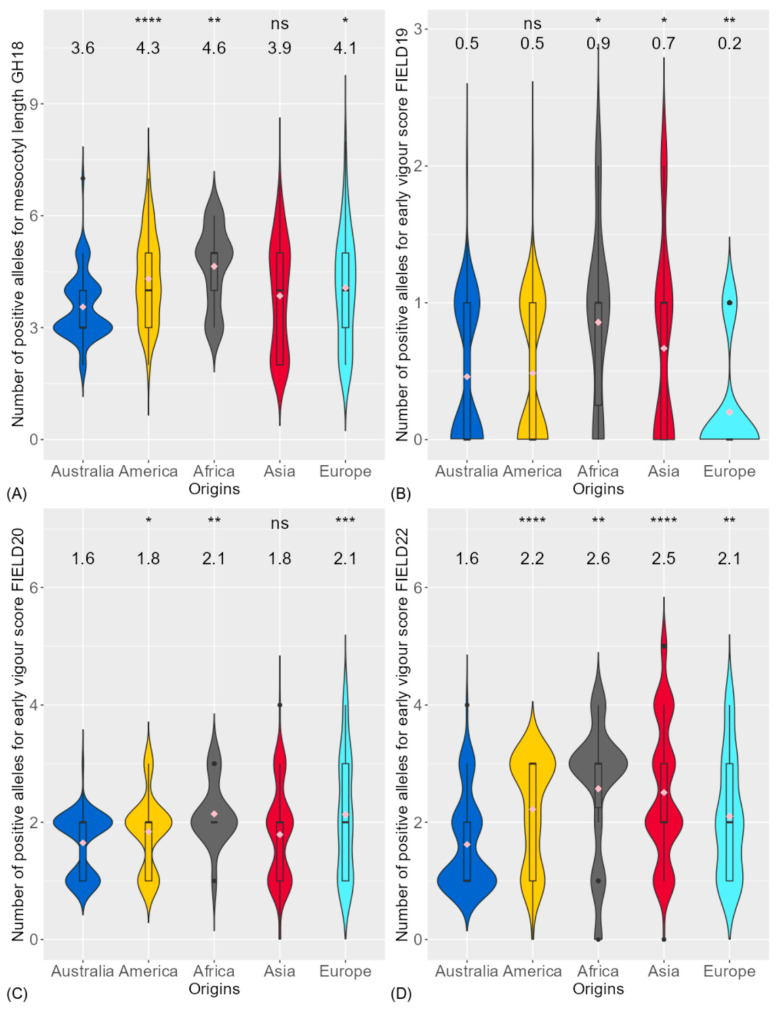
Comparison of genotypic origins for the number of positive alleles associated with (**A**) mesocotyl length in glasshouse experiment (GH18) and early vigour score in field experiment in (**B**) season 2018–2019 (FIELD19), (**C**) season 2019–2020 (FIELD20) and (**D**) season 2021–2022 (FIELD22) between Australia and different origins. Significance levels between Australian genotypes and other origins were displayed on the top of each figure: ns, not significant; *, *p* < 0.05; **, *p* < 0.01; ***, *p* < 0.001; ****, *p* < 0.0001. The average number of positive alleles of each origin were displayed below the significance level. The diamonds within each boxplot are the average number of positive alleles of each trait for each region of origin.

**Table 1 biology-13-00261-t001:** The mean, maximum and minimum values, heritability, least significant difference (LSD) in mesocotyl length, early vigour score, plant height, days to emergence, biomass per plant, biomass per m2 and light interception in two glasshouse experiments (GH18 and GH21) and three field experiments (FIELD19, FIELD20 and FIELD22).

Trait	Experiment	Mean	Maximum	Minimum	Heritability	LSD
Mesocotyl length (mm)	GH18	9.7	37.2	1.6	0.82	24.7
Early vigour score	FIELD19	4.8	9.2	0.9	0.57	2
	FIELD20	5.8	9.9	1.7	0.75	2.1
	FIELD22	4.7	8.2	0.8	0.65	2
Plant height (cm)	GH18	14.3	34.8	1.4	0.80	11.2
	FIELD20	29.6	47.0	12.8	0.82	6.7
	GH21	44.1	60.5	24.3	0.75	8.5
Days to emergence	GH18	10	16	4	0.42	5
	FIELD20	10	16	8	0.78	2
	GH21	5	10	3	0.40	2
Biomass per plant (mg)	GH18	12.1	32.5	0.7	0.75	10.0
Biomass per m^2^ (g m^−2^)	FIELD20	70.5	196.4	13.5	0.64	47.3
Light interception (%)	FIELD20	31	74	5	0.56	18

**Table 2 biology-13-00261-t002:** The quantitative trait loci (QTL, bold letters were QTL associated with multiple traits), single nucleotide polymorphism (SNP), chromosome (Chr.), position (Pos.), *p* value, minor allele frequency (MAF), SNP effect and phenotype variance explained (PVE) associated with days to heading (DTE), early vigour score (EVS), light interception (LI), biomass (mg), biomass m2 (g m^−2^), mesocotyl length (ML, cm) and plant height (PH, cm) in two aerobic glasshouse experiments (GH18, GH21) and three aerobic field experiments (FIELD19, FIELD20 and FIELD22) using the *japonica* diversity set.

QTL	Trait and Experiment	SNP	Chr.	Pos.	*p* Value	MAF	Effect	PVE
*qAEV1.1*	EVS FIELD19	1_6304441	1	6,304,441	1.91 × 10^−8^	0.36	−0.41	11.76
* **qAEV1.2** *	EVS FIELD22	1_32499722	1	32,499,722	9.70 × 10^−8^	0.13	−0.51	8.65
	EVS FIELD19	1_32510801	1	32,510,801	9.94 × 10^−7^	0.14	−0.42	21.44
	PH GH21	1_32510801	1	32,510,801	5.99 × 10^−9^	0.14	2.20	7.61
* **qAEV1.3** *	DTE FIELD20	1_34429355	1	34,429,355	5.06 × 10^−6^	0.14	0.62	9.73
	PH GH18	1_34600584	1	34,600,584	4.33 × 10^−9^	0.09	27.12	22.92
*qAEV1.4*	LI FIELD20	1_35998551	1	35,998,551	3.73 × 10^−8^	0.11	−0.06	7.29
* **qAEV1.5** *	EVS FIELD20	1_36358593	1	36,358,593	6.10 × 10^−8^	0.11	0.65	9.98
	ML GH18	1_36358593	1	36,358,593	8.48 × 10^−11^	0.11	−3.04	8.59
* **qAEV1.6** *	PH GH21	1_38847713	1	38,847,713	1.78 × 10^−10^	0.20	−2.23	7.61
	PH FIELD20	1_38847713	1	38,847,713	7.90 × 10^−8^	0.21	−18.71	5.01
*qAEV1.7*	Biomass GH18	1_40490610	1	40,490,610	7.44 × 10^−7^	0.11	−1.74	5.67
	Biomass GH18	1_40527461	1	40,527,461	1.96 × 10^−7^	0.41	−1.27	3.23
*qAEV1.8*	EVS FIELD20	1_42422994	1	42,422,994	3.68 × 10^−8^	0.06	−0.88	17.41
*qAEV2.1*	Biomass GH18	2_2713635	2	2,713,635	1.46 × 10^−6^	0.11	1.49	4.42
*qAEV2.2*	PH GH21	2_6543528	2	6,543,528	3.20 × 10^−7^	0.12	−1.99	6.01
*qAEV2.3*	DTE FIELD20	2_30871430	2	30,871,430	1.36 × 10^−6^	0.08	0.79	22.37
* **qAEV3.1** *	PH GH18	3_1248008	3	1,248,008	8.33 × 10^−11^	0.44	17.78	7.35
	Biomass GH18	3_1248008	3	1,248,008	5.00 × 10^−9^	0.44	1.30	3.42
*qAEV3.2*	DTE GH18	3_13599473	3	13,599,473	1.24 × 10^−8^	0.43	−0.66	8.74
*qAEV3.3*	ML GH18	3_15001141	3	15,001,141	2.72 × 10^−7^	0.47	1.59	1.77
*qAEV4.1*	Biomass GH18	4_4418061	4	4,418,061	7.89 × 10^−7^	0.39	1.17	2.71
*qAEV4.2*	ML GH18	4_9669965	4	9,669,965	3.08 × 10^−7^	0.07	3.75	5.82
* **qAEV4.3** *	LI FIELD20	4_12046452	4	12,046,452	5.27 × 10^−8^	0.48	0.04	8.94
	PH GH21	4_12046452	4	12,046,452	1.27 × 10^−8^	0.47	1.42	3.12
*qAEV4.4*	DTE GH18	4_21145795	4	21,145,795	3.79 × 10^−8^	0.17	0.94	9.23
*qAEV4.5*	EVS FIELD22	4_33718024	4	33,718,024	9.43 × 10^−7^	0.35	0.31	3.53
*qAEV5.1*	EVS FIELD22	5_14202497	5	14,202,497	9.73 × 10^−7^	0.48	−0.29	1.92
*qAEV5.2*	PH GH18	5_27968982	5	27,968,982	8.96 × 10^−11^	0.18	−23.12	1.44
* **qAEV6.1** *	PH FIELD20	6_21702460	6	21,702,460	7.16 × 10^−8^	0.07	33.18	36.33
	EVS FIELD22	6_21702460	6	21,702,460	1.10 × 10^−8^	0.07	−0.74	16.69
*qAEV6.2*	EVS FIELD22	6_23451697	6	23,451,697	1.01 × 10^−6^	0.19	0.38	4.19
*qAEV6.3*	LI FIELD20	6_27996796	6	27,996,796	3.48 × 10^−7^	0.06	0.07	34.84
*qAEV7.1*	EVS FIELD20	7_11559181	7	11,559,181	2.40 × 10^−6^	0.22	−0.38	2.42
*qAEV7.2*	DTE GH21	7_24657049	7	24,657,049	2.43 × 10^−7^	0.06	0.44	42.02
*qAEV7.3*	ML GH18	7_27838498	7	27,838,498	6.83 × 10^−6^	0.40	1.27	1.84
* **qAEV8** *	Biomass m^2^ FIELD20	8_10101994	8	10,101,994	5.07 × 10^−6^	0.33	−7.03	22.05
	EVS FIELD20	8_10101994	8	10,101,994	3.06 × 10^−6^	0.33	0.46	7.91
	ML GH18	8_10101994	8	10,101,994	1.44 × 10^−6^	0.32	−1.63	1.53
*qAEV9*	ML GH18	9_19288810	9	19,288,810	3.96 × 10^−6^	0.08	−2.40	7.19
*qAEV11.1*	ML GH18	11_4596939	11	4,596,939	2.05 × 10^−7^	0.35	1.38	1.33
*qAEV11.2*	PH FIELD20	11_21913577	11	21,913,577	4.17 × 10^−7^	0.06	−24.11	2.67
*qAEV11.3*	ML GH18	11_24666757	11	24,666,757	6.07 × 10^−6^	0.21	1.24	2.70

**Table 3 biology-13-00261-t003:** List of candidate genes functionally annotated in the representative quantitative trait loci (QTL).

QTL	Gene	Gene Identification	Gene Ontology Classification
*qAEV1.5*	*LOC_Os01g62460*	ZOS1-16-C_2_H_2_ zinc finger protein	Sequence-specific DNA-binding transcription factor activity; C2H2 zinc finger proteins have been shown to be involved in plant growth and development [57]
	*LOC_Os01g62480*	Laccase precursor protein	Drought tolerance, xylem structure, cell wall, cell length [58]
	*LOC_Os01g62500*	OsFtsH3 FtsH protease, homologue of AtFtsH3/10	Targets mitochondria and is involved in arginine metabolism during rice seed germination [59]
	*LOC_Os01g62514*	WRKY56	Sequence-specific DNA-binding transcription factor activity; WRKY gene family is involved in drought tolerance and root thickness [60]
	*LOC_Os01g62570*	ATP/GTP/Ca++ binding protein	Cell growth; post-embryonic development
	*LOC_Os01g62600*	Laccase precursor protein	Cell; response to stress; response to abiotic stimulus; leaf development under direct-sown and drought conditions [61]
	*LOC_Os01g62610*	Peptidyl-prolyl cis–trans isomerase, FKBP-type	FKBPs gene family regulates hormone signalling in plant growth and development, stress response and seed germination [62]
	*LOC_Os01g62630*	Aspartic proteinase nepenthesin precursor	Cell death; post-embryonic development; embryo development; drought response [63]
	*LOC_Os01g62660*	MYB family transcription factor	Sequence-specific DNA-binding transcription factor activity; drought response [64]
	*LOC_Os01g62760*	Protein phosphatase 2C	Drought response; protein phosphatase 2C involved in ABA metabolism [65]
	*LOC_Os01g62800*	Methyltransferase	Methyltransferase gene family is related to the seed vigour index [66]
	*LOC_Os01g62810*	Regulator of chromosome condensation	Regulation of plant organ elongation [67]
	*LOC_Os01g62840*	Mannose-1-phosphate guanyltransferase	Drought response [63]; mannose-1-phosphate guanyltransferase regulates chlorophyll retention and seedling growth [68]
	*LOC_Os01g62900*	Amino acid kinase	Drought response [69]
	*LOC_Os01g62920*	Homeodomain protein	Sequence-specific DNA-binding transcription factor activity; homeodomain protein involved in plant hormone regulation including auxin, ABA, cytokinin and GA [23]
	*LOC_Os01g62950*	RAS-related protein	Drought response [70]; cellular component; RAS-related protein responses to ABA, lipid metabolic processes and carbohydrate metabolic processes [71]
	*LOC_Os01g63010*	Universal stress protein domain containing protein	Regulates genes under direct-sown and drought stress conditions [61]
	*LOC_Os01g63060*	Phosphatidic acid phosphatase-related	Regulates the plant height, growth and development of rice [72]
*qAEV8*	*LOC_Os08g16260*	Cytochrome P450 protein	Cytochrome P450 protein regulates cell size in the embryo and apoptosis in the endosperm [73]
	*LOC_Os08g16320*	Cytochrome P450 protein	Cytochrome P450 protein regulates cell size in the embryo and apoptosis in the endosperm [73]
	*LOC_Os08g16480*	ATPase, AFG1 family domain-containing protein	ATPase is involved in germination percentage, seedling growth in terms of root and shoot lengths [74]
	*LOC_Os08g16570*	Expressed protein	Drought resistance [75]
	*LOC_Os08g16600*	WD-40 repeat protein family, expressed	WD-40 gene family is involved in signal transduction and hormone-controlled plant cell division [68]

## Data Availability

Data is contained within the article or Appendix A.

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
