# Peer review of "Genome-Wide Association Study of Early Vigour-Related Traits for a Rice (Oryza sativa L.) japonica Diversity Set Grown in Aerobic Conditions"

_biology, 2024, doi:10.3390/biology13040261_

Round 1

Reviewer 1 Report

Comments and Suggestions for Authors

The study identified QTLs and candidate genes for early vigour-related traits in aerobic condition using a japonica rice diversity set through GWAS analysis. Five experiments were conducted in field and greenhouse across multiple environments. A slight drawback is that the traits investigated in each experiment were not uniform.

Here are some minor comments:

1, “Oryza sativa” should be in italic. Please modify it in the whole MS, including reference section.

2, Line 86, “Zhang, Yu, Yu, Huang and Zhu” could be “Zhang, et al.”

3, Line 104, please delete “and plant materials”.

4, Fig1, “0.2” was missing on X-axis. Additionally, some names were overlapped, and hard to be recognized.

5, Fig2, the data of EVS19 was not seen. Since EVS were carried out across three years, why not show all the data?

6, Line 305, “p” should be in italic.

7, Line 331, here “except Chr. 10”, but from Table2, no QTL was detected on Chr. 12 also.

8, How the threshold was determined? As shown in Table2, several QTLs demonstrated very low PVE. For example, the p value of qAEV8 for MLGH18 was 1.44E-0.6 and the PVE was 1.53%.

Comments on the Quality of English Language

Minor editing of English language required

Author Response

The study identified QTLs and candidate genes for early vigour-related traits in aerobic condition using a japonica rice diversity set through GWAS analysis. Five experiments were conducted in field and greenhouse across multiple environments. A slight drawback is that the traits investigated in each experiment were not uniform.

Here are some minor comments:

1, “Oryza sativa” should be in italic. Please modify it in the whole MS, including reference section.

We thank the reviewer for the notification and corrected them in the manuscript including reference.

Changed to “Oryza sativa” in the manuscript at line 3, line 228 and reference section.

2, Line 86, “Zhang, Yu, Yu, Huang and Zhu” could be “Zhang, et al.”

We thank the reviewerfor the notification and corrected them in the manuscript.

Changed to “Zhang, et al” in line 86.

3, Line 104, please delete “and plant materials”.

We thank the reviewer for the notification and removed this in the manuscript. “and plant materials” was deleted at line 104.

4, Fig1, “0.2” was missing on X-axis. Additionally, some names were overlapped, and hard to be recognized.

We thank the reviewer for the notification and corrected them in the manuscript.

The 0.2 and -0.2 are added on X-axis. Spacing around names was modified.

5, Fig2, the data of EVS19 was not seen. Since EVS were carried out across three years, why not show all the data?

We thank for the reviewer’s suggestion and add the EVS19 in the manuscript. The boxplot of EVS19 was added in Fig 2 at line 308 and Figure 6 at line 387.

6, Line 305, “p” should be in italic.

We thank the reviewer for the comments and corrected them in the manuscript. All “p” were changed to italic format at line 313, line 392-393.

7, Line 331, here “except Chr. 10”, but from Table2, no QTL was detected on Chr. 12 also.

We thank the reviewer for the comments and corrected them in the manuscript. Added “chromosome 12” after “except chromosome 10” at line 339.

8, How the threshold was determined? As shown in Table2, several QTLs demonstrated very low PVE. For example, the p value of qAEV8 for MLGH18 was 1.44E-0.6 and the PVE was 1.53%.

We followed the GAPIT standard according to “https://rdrr.io/github/jiabowang/GAPIT3/src/R/GAPIT.R” in R studio. We used “cutoff=0.05” for the threshold for significant, and “Random.model = TRUE” which ran random model to estimate PVE values for significant markers after GWAS. The SNPs with false discovery rate < 0.05 was set as the threshold to be considered significant. According to Wang and Zhang [1], “In GAPIT3, the percentage of total phenotypic variance explained (PVE) by significantly associated markers (P values < Bonferroni threshold) is evaluated. A Bonferroni multiple test threshold is used to determine significance. The associated markers are fitted as random effects in a multiple random variable model. The model also include other fixed effects that are used in GWAS to select the associated markers. The multiple random variable model is analyzed using an R package, lme4, to estimate the variance of residuals and the variance of the associated markers. The percentage explained by the markers are calculated as their corresponding variance divided by the total variance, which is the sum of residual variance and the variance of the associated markers.”. Moreover, Wang and Zhang [1] mentioned “The percentage of PVE by a marker is correlated with its MAF and magnitude of marker effect.” In their study. Using the mesocotyl length SNP in the current study as example, the PVE was correlated MAF and marker effect which is consistent with the results in Wang and Zhang [1]’s study.

No changes were made to the manuscript.

[1] Wang, J.; Zhang, Z. GAPIT Version 3: Boosting Power and Accuracy for Genomic Association and Prediction. Genomics, Proteomics & Bioinformatics 2021, 19, 629-640, doi:https://doi.org/10.1016/j.gpb.2021.08.005.

Reviewer 2 Report

Comments and Suggestions for Authors

The MS used 302 rice genotypes to identify the QTLs and candidate genes associated with early vigour related traits in aerobic conditions. Totally 32 QTL were identified and 23 candidate genes were found in two important QTL regions ( qAEV1.5 and qAEV8 ). The research topic aiming to the rice production at aerobic condition which is meaningful. Two glasshouse and three field experiments were conducted which guarantee the phenotype data more accurate. However, two information should be provided before considering acceptance.

1) The soil water condition should be provided in each experiment, such as soil water potential. Since the significant interaction between genotype and environment,  different QTLs could be identified under different water conditions.

2)  23 candidate genes were proposed, some basic verification work should be conducted on important genes, such as qRT-PCR.

3) Other small suggestion: the written description should be concise, such as in part 2.1,   University of Queensland, Australia occured three times in one paragraph, which can be improved and concise more.\

4) The Plant materialsoccured in both titles of parts 2.1 and 2.2, please check.

5) In result: heritability was mentioned, but no relative description on it was found in method part, pls supplement relevant contents.

6) In line 86 and line 467, I dont think it is necessary to mentioned all anthors names in text when citing references.

 The writting should be improved more.

Author Response

The MS used 302 rice genotypes to identify the QTLs and candidate genes associated with early vigour related traits in aerobic conditions. Totally 32 QTL were identified and 23 candidate genes were found in two important QTL regions (qAEV1.5 and qAEV8 ). The research topic aiming to the rice production at aerobic condition which is meaningful. Two glasshouse and three field experiments were conducted which guarantee the phenotype data more accurate. However, two information should be provided before considering acceptance.

1) The soil water condition should be provided in each experiment, such as soil water potential. Since the significant interaction between genotype and environment, different QTLs could be identified under different water conditions.

We thank the reviewer for the suggestion. However, the soil water condition was not recorded in the current experiments. We thank for the reviewer’s suggestion and will measure the soil water potential in future experiments.

No changes were made.

2)  23 candidate genes were proposed, some basic verification work should be conducted on important genes, such as qRT-PCR.

We thank the reviewer for the suggestion. However, this was not the objective of this study. Also, “validation of the identified QTL and candidate genes in other genetic backgrounds is crucial to ensure the stability and reliability before adopting it into the rice breeding programs.” was mentioned in the discussion.

No changes were made.

3) Other small suggestion: the written description should be concise, such as in part 2.1,   “University of Queensland, Australia” occured three times in one paragraph, which can be improved and concise more.

We thank the reviewer for the comments and simplified the manuscript.

Added “at the University of Queensland, Australia” after “Two glasshouse experiments and three field experiments were conducted” at line 105-106. “the University of Queensland, Australia” was removed from the rest of the paragraph from line 105-113.

4) The “Plant materials”occured in both titles of parts 2.1 and 2.2, please check.

We thank the reviewer for the notification and removed them in the manuscript.

The “plant materials” was removed at line 104.

5) In result: heritability was mentioned, but no relative description on it was found in method part, pls supplement relevant contents.

We thank the reviewer for the notification and corrected them in the manuscript.

Added the description of heritability in part 2.6 from line 206-211.

6) In line 86 and line 467, I don’t think it is necessary to mentioned all anthors’ names in text when citing references.

We thank the reviewer for the notification and corrected them in the manuscript.

Changed to “Zhang, et al” in line 86, and “Wu, et al” in line 476